# A Paratope-Enhanced Method to Determine Breadth and Depth TCR Clonal Metrics of the Private Human T-Cell Vaccine Response after SARS-CoV-2 Vaccination

**DOI:** 10.3390/ijms241814223

**Published:** 2023-09-18

**Authors:** Dalin Li, Ana Jimena Pavlovitch-Bedzyk, Joseph E. Ebinger, Abdul Khan, Mohamed Hamideh, Akil Merchant, Jane C. Figueiredo, Susan Cheng, Mark M. Davis, Dermot P. B. McGovern, Gil Y. Melmed, Alexander M. Xu, Jonathan Braun

**Affiliations:** 1Inflammatory Bowel Disease Institute, Department of Medicine, Cedars-Sinai Medical Center, Los Angeles, CA 90048, USA; dalin.li@cshs.org (D.L.); abdul.khan@cshs.org (A.K.); mhamideh31@gmail.com (M.H.); dermot.mcgovern@cshs.org (D.P.B.M.); gil.melmed@cshs.org (G.Y.M.); 2Computational and Systems Immunology Program, Stanford University School of Medicine, Stanford, CA 94305, USA; pavlo@stanford.edu (A.J.P.-B.); mdavis@cmgm.stanford.edu (M.M.D.); 3Department of Cardiology, Smidt Heart Institute, Cedars-Sinai Medical Center, Los Angeles, CA 90048, USA; joseph.ebinger@cshs.org (J.E.E.); susan.cheng@cshs.org (S.C.); 4Cedars-Sinai Cancer and Department of Medicine, Cedars-Sinai Medical Center, Los Angeles, CA 90048, USA; akil.merchant@cshs.org (A.M.); jane.figueiredo@cshs.org (J.C.F.); alexander.xu@cshs.org (A.M.X.); 5Department of Microbiology and Immunology, Howard Hughes Medical Institute, Stanford University School of Medicine, Stanford, CA 94305, USA

**Keywords:** T-cell receptors, T-cells, SARS-CoV-2, CDR3 domains, GLIPH

## Abstract

Quantitative metrics for vaccine-induced T-cell responses are an important need for developing correlates of protection and their use in vaccine-based medical management and population health. Molecular TCR analysis is an appealing strategy but currently requires a targeted methodology involving complex integration of ex vivo data (antigen-specific functional T-cell cytokine responses and TCR molecular responses) that uncover only public antigen-specific metrics. Here, we describe an untargeted private TCR method that measures breadth and depth metrics of the T-cell response to vaccine challenge using a simple pre- and post-vaccine subject sampling, TCR immunoseq analysis, and a bioinformatic approach using self-organizing maps and GLIPH2. Among 515 subjects undergoing SARS-CoV-2 mRNA vaccination, we found that breadth and depth metrics were moderately correlated between the targeted public TCR response and untargeted private TCR response methods. The untargeted private TCR method was sufficiently sensitive to distinguish subgroups of potential clinical significance also observed using public TCR methods (the reduced T-cell vaccine response with age and the paradoxically elevated T-cell vaccine response of patients on anti-TNF immunotherapy). These observations suggest the promise of this untargeted private TCR method to produce T-cell vaccine-response metrics in an antigen-agnostic and individual-autonomous context.

## 1. Introduction

The COVID-19 pandemic has spotlighted the importance of the human immune response to viral immunization. New mRNA and other biologic immunization platforms elicit distinct responses across immunity, and antibody-based measurements are commonly used for assessment of vaccine efficacy, coverage for new variants, quantitative durability of immunity, and resultant recommendations for timing of re-immunization [1]. Several considerations point to the independent value of T-cell responses as a complement to antibodies in the assessment of anti-viral immunity. In general, the relative contribution of antibody and T-cell effector functions in viral immunity is characteristic for each pathogen [2,3]. For SARS-CoV-2, antibodies play a direct role in transmission protection, whereas T-cell responses support antibody formation and attenuate disease severity after viral transmission. While the durability of antibody and T-cell immunity for SARS-CoV-2 temporally correlate [4,5], CD8+ T-cell levels are distinctly prolonged [6,7,8]. Unlike antibody immunity, T-cells cross-respond to Omicron, delta, and beta variants at comparable magnitudes [9,10,11]. Also, whereas antibody immunity is reduced by anti-TNF and preserved by mycophenolate immunosuppressant therapy, T-cell responses are augmented by anti-TNF and reduced by mycophenolate [11,12,13,14,15,16]. Thus, a more robust understanding of the dynamics and effectors of T-cell immunity, independent of antibody immunity, will be essential to mobilizing every aspect of adaptive immunity. This is especially true for the current phase of the COVID-19 pandemic, where healthy patients with normal antibody and T-cell immunity fare well and treatment focus is now shifting to specific vulnerable populations with altered immunity, such as cancer patients who receive B-cell ablative therapies and thus cannot generate antibodies effectively [11].

T-cell responses may be measured by ex vivo biologic assessment of antigen-induced cytokine responses or TCR clonal analysis of blood-derived DNA. Whereas the former approach is most commonly employed, there is rising interest in TCR-based molecular analysis due to the minimal requirements for cell handling, suitability for use on common archival biospecimens, and advances in analytic methods and their commercial dissemination [17,18,19]. Quantitatively, the virus specificity of a patient’s TCR repertoire can be characterized by breadth (a metric for the number of unique virus-specific clones among the total population of clones) and depth (a metric for the number of virus-specific T-cells among the total population of cells) [19,20,21]. Conceptually, distinguishing TCR-based antigen-specific responses from the nonspecific TCR repertoire can be done using strategies based on enumerating temporal clonal dynamics or on shared cognate structures required for antigen-binding [8,19,22,23,24].

The temporal clonal dynamics of antigen-specific TCR clones follow a pattern of selective expansion, relative to clones without antigenic specificity, in the early time period after antigenic challenge. For example, after viral antigen immunization, a pre-study cohort with known infection status is typically analyzed, and the extent of TCR sharing between a sample and positive controls of the pre-study cohort is measured. However, this approach only identifies TCR clones widely utilized across individuals (public specificities) [25,26,27] that represent a very small fraction of the total clonal response for each individual (private specificities) and are further restricted by the genetic representation of HLA alleles in the pre-study cohort when used to subsequently analyze TCR responses in genetically divergent individuals [18,22,23,26]. In this study, we have developed a Poisson-based analysis to identify candidate vaccine-specific TCRs by temporal clonal dynamics.

Cognate structure analysis takes advantage of the concept that antigen-binding TCR structures (paratopes) will likely be homologous among TCRs sharing a particular antigen-binding specificity. The identification of such cognate structures is a challenging and rapidly developing computational field. Some strategies are structure-agnostic, using a range of machine-learning methods to uncover TCR (particularly complementary-determining region 3, CDR3) features distinguishing individuals differing by naïve-versus-experienced immune states. While these strategies may be powered by large sample sizes and sophisticated computational methods, they are limited by the infrequency of antigen-specific clones (typically 10^−6^) and the dependence on public specificities [19]. Other strategies directly focus on uncovering cognate CDR3 features for target peptide-MHC structures. This direct approach is challenging because of the great diversity of CDR3 features used by functionally validated T-cell clones, but progress has been made using protein sequence alignment, peptide similarity, and shared motifs, among other methods [19,28,29]. In this study, we have adopted the GLIPH2 shared-motif identification algorithm to identify candidate vaccine-specific TCRs by shared cognate structures for antigen-binding. The GLIPH2 algorithm groups TCR sequences by their shared peptide-MHC specificity (motifs). Using an input TCR dataset from a cohort, the algorithm outputs a parameter-rich output of findings, including significant motif and TCR “convergence” lists, and a collection of scores for enrichment of motif, V-gene, CDR3 length, shared HLA, and proliferation count [26,28].

The present study presents an approach to define private specificities in the SARS-CoV-2 vaccine response. To do so, we identified responder TCR clones in individuals by two independent and complementary criteria. First, with regard to clonal dynamics, we applied a Poisson distribution of pairwise comparison of TCRs across different time points before and after vaccination. Second, we created self-organizing maps of clonal responders and applied GLIPH2 to test for preserved CDR3 structural features of responder TCR clones. For both, using clones significantly associated with the vaccine response, we calculated aggregate depth and breadth metrics of private T-cell vaccine response. Our findings indicate that the metrics of the private TCR response robustly recapitulated that of the public TCR response. The GLIPH2 analysis provided evidence that responder TCR clones displayed structural features relevant to the S-antigen vaccine immunogen. These findings suggest the potential for this approach to define metrics of the private TCR response to vaccination. 

## 2. Results

### 2.1. Cohorts

Detailed demographic information of the IBD and HCW cohorts has been previously reported [15,16], with the demographics of the two cohorts summarized in Table 1. Individuals in these cohorts were studied at up to four time points in relation to SARS-CoV-2 mRNA vaccination. As summarized in Table 1, a total of 927 samples from 515 individuals were included in the current study, of which 584 samples were from the IBD cohort and 343 from the HCW cohort. A total of 39.08% of the samples were from females and 73.39% received the Pfizer vaccine. In IBD patients, 34.59% of patients received anti-TNF treatments and 76.54% received at least one biologic. 

### 2.2. Pipeline

The pipeline for the untargeted private TCR vaccine-response method is summarized in Figure 1. Subjects were sampled at up to four time points in relation to a two-dose mRNA SARS-CoV-2 vaccination regimen. TCR β chains were sequenced using the Adaptive Biotechnologies Immunoseq platform. Putative SARS-CoV-2 private vaccine-response TCRs for each individual were identified by log-linear regression. A SOM analysis of putative TCRs was used to refine TCRs for likely trajectories of vaccine response. GLIPH2 analysis was performed on antigen-associated TCRs from the ex vivo MIRA SARS-CoV-2 database, and putative private TCRs were refined for those sharing a GLIPH group with a specific MIRA epitope. TCRs found in both the SOM and GLIPH lists were selected as high-likelihood private TCR vaccine-response candidates.

### 2.3. Filtering by Temporal Self-Organizing Maps and GLIPH2 Structurally Cognate TCRs

We used SOMs to separate individual TCRs by their prototypic trajectories over time in patients with fully available time courses, uncovering 48 SOMs for further analysis (Figure 2A,B). From these candidates, we identified 3 or 10 SOMs with likely vaccination-specific trajectories that were selected for further analysis (termed Top 3 and Top 10 nodes, respectively; see Methods), where the Top 10 node candidate TCRs were subjected to an additional condition of appearing selectively in Top 10 trajectory nodes. CDR3 trajectories populating a representative vaccine candidate SOM node are detailed in Figure 2C. 

As a second filter, we used GLIPH2 to identify TCRs with sequence similarity to TCRs from a set of ~130,000 putative SARS-CoV-2-specific TCRs (MIRA). From this reference dataset, GLIPH analysis produced 4599 vaccine candidate CDR3s, and these CDR3s overlapped with candidate CDR3s in Top 3 nodes and Top 10 nodes (Figure 2D). The log_10_ frequency of CDR3s in each SOM node is shown in Figure 2E. The percentage of CDR3s in each SOM node that shared GLIPH groups with MIRA CDR3s was enriched in the likely vaccine trajectories (Figure 2F). For brevity, we term the TCRs found in the overlap of these SOM and GLIPH2 filters as TCR1 (for Top3GLIPH) and TCR2 (for Top10GLIPH). 

### 2.4. The Private TCR Vaccine Response and Its Relation to Vaccine and Clinical Features

We first produced breadth and depth TCR response metrics for the Top 3 or Top 10 TCRs. However, no significant association with vaccine time course or clinical variables was observed for subjects including all their matching TCRs for Top 3 or Top 10 SOMs nor with TCRs identified by the GLIPH filter alone (Appendix A). 

We therefore produced breadth and depth metrics for the private TCR vaccine response of individuals using double-filtered TCRs (either TCR1 or TCR2) and compared them with the breadth and depth metrics produced by a targeted public TCR response method (Adaptive Biotechnologies) from the same subjects and TCR datasets [15,16]. The intra-individual correlations of the private and public TCR vaccine responses are shown in Figure 3. The correlations for breadth were negligible or weak (r = 0.06 and 0.15, *p*-value 0.09 and 6.20 × 10^−8^, respectively, for TCR1 and TCR2), while the correlations for depth were slightly stronger (r = 0.28 and 0.25, *p*-value 6.10 × 10^−31^ and 2.11 × 10^−28^, respectively, for TCR1 and TCR2).

We then tested the time variation of these breadth and depth metrics in the IBD and HCW cohorts (Table 2 and Figure 4). For both groups of TCRs, we observed strong differences in the IBD cohort for comparisons of week2 vs. baseline (*p* = 10^−10^ to 10^−23^) and week8 vs. baseline (*p* = 10^−9^ to 10^−17^). Similar findings were also observed in the HCW cohort. For example, increased breadth of TCR2 was observed in week8 compared with baseline, with *p*-values of 2.05 × 10^−14^ and 2.41 × 10^−10^ (estimate = 0.29 and 0.20), respectively, for IBD and HCW.

The association between TCR metrics and subject age in the IBD and HCW cohorts is shown in Table 3. Consistent with the trend in public vaccine-response TCR metrics in previous analyses [15,16], higher age is associated with lower depth but not with breadth in both IBD and HCW, e.g., for TCR2 depth, the association *p*-value is 5.51 × 10^−3^ in IBD (estimate = −0.004) and 0.039 in HCW (estimate = −0.0038), with a combined *p*-value of 5.59 × 10^−4^ (estimate = −0.0039) in all the subjects.

The difference of TCR metrics in IBD patients with anti-TNF treatments compared with those without anti-TNF is shown in Table 4. Patients on corticosteroids were excluded due to their effect on T-cell responses. We observed marginally higher depth for TCR1 at 2 weeks after the 2nd dose (estimate = 0.12, *p* = 0.014) and a trend in this direction at 8 weeks (estimate = 0.08, *p* = 0.074). We also performed these anti-TNF comparisons using the Mann–Whitney U test, yielding consistent results (*p* = 0.027 and 0.089, respectively, for the above-mentioned time points). No statistically significant association was observed for TCR2 depth or any breadth. 

## 3. Discussion

The primary result of our study is a three-tiered approach to defining private TCR repertoires using Poisson statistics, trajectories, and GLIPH. Each filter tier addresses a distinct aspect of the private TCR response to vaccination against COVID-19. First, vaccination should active T-cell response and induce the clonal expansion of the activated T-cells, which can be identified via the Poisson statistic and the trajectory filters. Second, the expansion should be specific to vaccination and not a concurrent, unrelated response. The trajectories address this by defining the time scale of an expected expansion and filtering out sequences that do not respond consistently across patients. Finally, vaccine-specific TCRs must recognize viral antigens presented on major histocompatibility complex (MHC) via biomolecular interactions between the TCR sequence and viral peptide. This requirement limits the CDR3 peptide sequence in ways that are challenging to predict, and different CDR3 sequences have been found to bind the same antigen, likely dependent on whether similar chemical motifs are presented [29]. The GLIPH filter addresses this biomolecular recognition aspect without requiring a direct matching sequence [20]. 

The heterogeneity of TCR CDR3 sequences is apparent in our data, as over 10M unique sequences were sequenced. Our filters generate a “private”, de novo reference list in the sense that these sequences are not required to have been previously observed but instead “look” (by sequence via GLIPH) and “act” (by trajectory via SOM) like vaccine-specific TCRs. Omitting any of these filters removes the ability of the partially filtered CDR3s to discriminate the time course of vaccination. Using our strategy requires the sequence information of existing, putative antigen-specific CDR3s. Here, we do not treat these sequences as a reference list, and instead, our GLIPH and trajectory filters proactively remove sequences that do not appear consistent with vaccination. The list of MIRA CDR3s used in our GLIPH filter is not stringent by independent validation, and so, our strategy simply uses MIRA CDR3s as a guide to detect novel CDR3s in a way that is generalizable to other putative antigen-specific CDR3s. 

We note that there are limitations to this private TCR vaccine-response method and its clinical correlations. We did not observe a strong correlation between our private sequences and a public strategy, as evidenced by the lack of correlation between breadth metrics in Figure 3, which focuses on the number of unique antigen-specific TCRs. However, the greater correlation of depth, which incorporates the functional expansion of TCRs, suggests that our private strategy and the public strategy begin to converge on functional performance-dependent metrics. Finally, we note that this population is limited in ethnic and racial diversity, so validation of these findings in cohorts of distinct ethnic and racial composition will be important.

## 4. Methods and Materials

### 4.1. Study Subjects and Their T-Cell Clonal Composition

This paper analyzes data from two independent study cohorts, one consisting solely of inflammatory bowel disease (IBD) patients and one consisting of healthcare workers. Details of the two cohorts were previously reported [15,16]. The Coronavirus Risk Associations and Longitudinal Evaluation-IBD study cohort (here termed the “IBD” cohort) enrolled adults with IBD under care for IBD at the Cedars-Sinai Medical Center who were planning to receive or who already received the SARS-CoV-2 vaccination (January to July 2021). Study participants underwent a 2-dose series of either the BNT162b2 (Pfizer-BioNTech, New York City, NY, USA) or mRNA-1273 (Moderna, Cambridge, MA, USA) vaccines. Participants completed baseline surveys detailing medical history at the time of vaccination, including age, sex, IBD classification, vaccine type, and medication classes. Participant blood collections occurred after dose 1 (from 5 days after dose 1 until the day of dose 2); after dose 2 (from 2 to 13 days after dose 2); and at 2 weeks (14 to 29 days), 8 weeks (30 to 84 days), and 16 weeks (85 to 140 days) after dose 2. Participants were excluded if they received the Ad26.COV2 vaccine (Johnson & Johnson New Brunswick, NJ, USA), had prior COVID-19 (a positive SARS-CoV-2 nucleocapsid IgG result at any time point), or did not receive both mRNA doses. 

The healthcare worker study cohort (here termed the “HCW” cohort) was a longitudinal cohort study of healthcare workers who received vaccination with Pfizer-BioNTech (BNT162b2). Beginning on 11 May 2020, this study enrolled 6318 active employees working in the Cedars-Sinai Health System, located in Los Angeles County, California. The Cedars-Sinai organization includes two hospitals (Cedars-Sinai Medical Center and Marina del Rey Hospital) in addition to multiple clinics in the Cedars-Sinai Medical Delivery Network. All active employees (totaling n~15,000) were invited to participate in the study by providing peripheral venous blood samples under the same collection schedule as described for the IBD cohort. Participants completed surveys on medical history, exposures, and symptoms at baseline and at serial time points over the course of the study. History of SARS-CoV-2 infection prior to vaccination was determined based on self-reporting or positive CoV-2 nucleocapsid IgG. Participants in this cohort were excluded if they received a vaccine other than BNT162b2, did not complete the two initial vaccine doses, had evidence of a SARS-CoV-2 infection, or their infection status could not be confirmed.

### 4.2. Untargeted Private TCR Vaccine-Response Method

T-cell clonal composition in blood DNA of study subjects was quantified by T-cell receptor (TCR) β sequencing of blood genomic DNA (ImmunoSEQ, Adaptive Biotechnologies, Seattle, WA, USA) as previously described [15,16]. The method pipeline is summarized in Figure 1. First, a log-linear model analysis based on a Poisson distribution was utilized to perform a pairwise comparison of counts of each TCR across different time points in each individual. TCRs with significant *p*-values after Bonferroni’s correction for multiple testing were combined into a list of individual subject (private) candidate TCRs.

Second, self-organizing maps (SOMs) were generated using the kohonen package in R (https://cran.r-project.org/web/packages/kohonen/index.html, accessed on 13 July 2023). CDR3 frequencies from patients with 4 time points (n = 26) were collected, and data were transformed by adding the minimum observed frequency (10^−6^) and log-transforming. The trajectories of every detected CDR3 in every participant were used to form a SOM with 48 nodes. Each node represents a group of CDR3s with similar trajectory, and vaccine candidate nodes were selected manually. The same CDR3s were found across nodes due to different trajectories observed in one participant versus another. Individual CDR3s were selected as likely vaccine candidates if they appeared in at least one participant (Top 3 SOM nodes), or if they appeared in at least one participant, and at least 50% of their appearances were in a vaccine candidate node (Top 10 SOM nodes). 

Third, the GLIPH2 algorithm [26,27,28] was used to analyze two CDR3 datasets: (a) sequences identified by the untargeted method that were significantly associated with the vaccine response after Bonferroni correction (*p* < 1.01 × 10^−9^) and (b) putative SARS-CoV-2-specific CDR3s from the MIRA database [19], which is a large-scale database of T-cell receptor beta (TCR β) sequences for response to the SARS-CoV-2 virus. CDR3s from patients were denoted as vaccine candidates if they were found in a GLIPH group with at least 5 MIRA CDR3s and at least half of the MIRA CDR3s were assigned to the same SARS-CoV-2 epitope, to remove noise and non-specific candidates, respectively. 

### 4.3. Breadth and Depth Metrics and Feature Analysis of the Private TCR Response

After the double filtering by self-organizing maps and GLIPH2, breadth and depth based on the private TCR lists were calculated using the approach in Snyder et al. [19] for each sample included in the analysis. We termed this the untargeted private TCR method. Briefly, the breadth is defined as the fraction of unique virus-specific TCR clones, and the depth is defined as the relative expansion of virus-specific TCR clones. For both metrics, larger values indicate more robust virus-specific TCR responses.

The impact of demographic and clinical features on the breadth and depth of the private TCRs at different time points was evaluated using a generalized linear model (GLM). That is, after an inverse normal transformation (using the RNOmi package in R) (https://cran.r-project.org/web/packages/RNOmni/index.html, accessed on 13 July 2023), the GLM was adjusted for cohort, age, TNF medication, and sex, as appropriate. Differences of TCR metrics between different time points were examined using a linear mixed model (LMM) to account for partial overlap of the subjects in different time points (using the lme4 package in R) (https://cran.r-project.org/web/packages/lme4/index.html, accessed on 13 July 2023). Stratified analysis by cohorts was also performed since the private TCR list was generated solely based on the IBD cohort. Correlation among the breadth and depth metrics was performed using the Spearman correlation due to data skewness. Results were considered significant where *p* < 0.05.

## 5. Conclusions

This study demonstrates that a paratope-enhanced method for detection of private TCR vaccine responses is significant TCR responses by two independent criteria: those selected by functional validation via peptide stimulation (MIRA candidates) and those observed via clinical trends and vaccination timing (Table 2), age (Table 3), and anti-TNF response (Table 4).

## Figures and Tables

**Figure 1 ijms-24-14223-f001:**
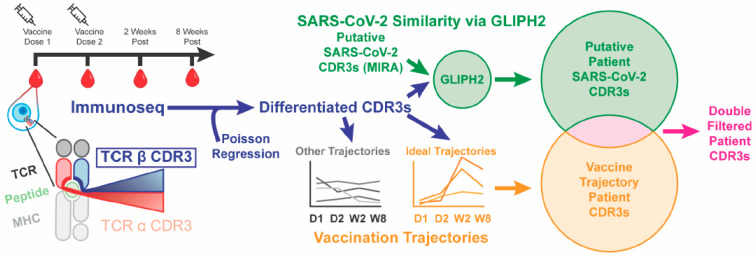
Pipeline for determining the untargeted private TCR vaccine response. TCR β chains were sequenced using the Adaptive Biotechnologies Immunoseq platform at up to 4 time points. Putative SARS-CoV-2 TCRs were identified by performing a Poisson regression. After combining the GLIPH2 analysis with putative TCRs and TCRs from the MIRA database, putative TCRs were refined for those sharing a GLIPH group with a specific MIRA epitope. A SOM analysis of putative TCRs was also used to refine TCRs for likely trajectories of vaccine response. TCRs found in both refined lists were the most likely private TCR vaccine-response candidates.

**Figure 2 ijms-24-14223-f002:**
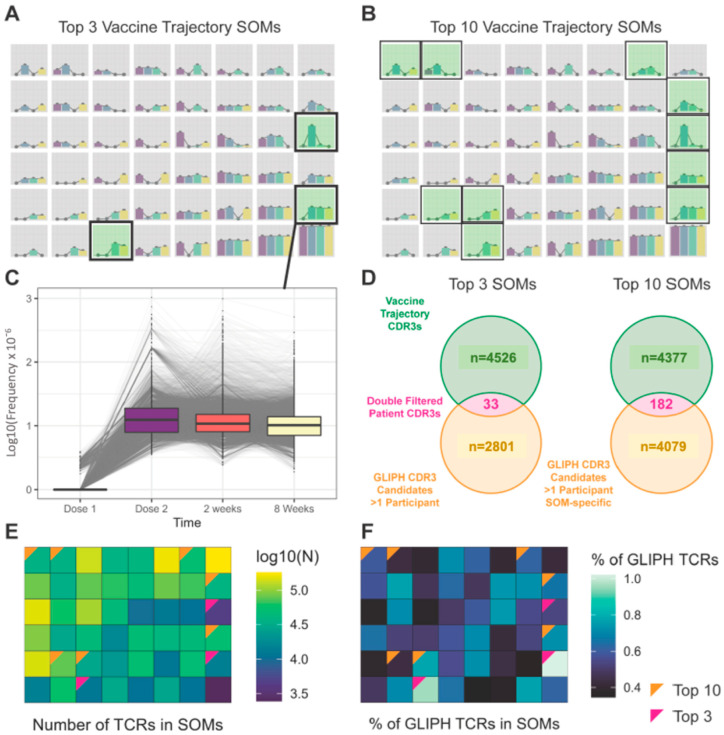
Self-organizing maps of candidate private TCR vaccine-response clones. TCRs formed 48 SOM nodes defining prototypical trajectories (log-scaled). Either 3 (**A**) or 10 (**B**) candidate nodes were chosen as vaccine candidates, which increased after vaccination (green highlighted). These were termed Top 3 and Top 10 nodes, respectively. (**C**) CDR3 trajectories populating a vaccine candidate SOM node are shown in detail. (**D**) For GLIPH analysis of Top 3 nodes, all CDR3s that appeared in at least one participant were selected (n = 2834), with 33 appearing in both GLIPH- and SOM-refined lists. For Top 10 nodes, a distinct specificity filter was applied, resulting in 4261 CDR3s and 182 in both lists. Due to overlap in unique CDR3s in the two nodes, GLIPH produced a total of 4599 vaccine candidate CDR3s. (**E**) The log_10_ frequency of CDR3s in each SOM node. (**F**) The percentage of CDR3s in each SOM node that shared GLIPH groups with MIRA CDR3s was enriched in the likely vaccine trajectories.

**Figure 3 ijms-24-14223-f003:**
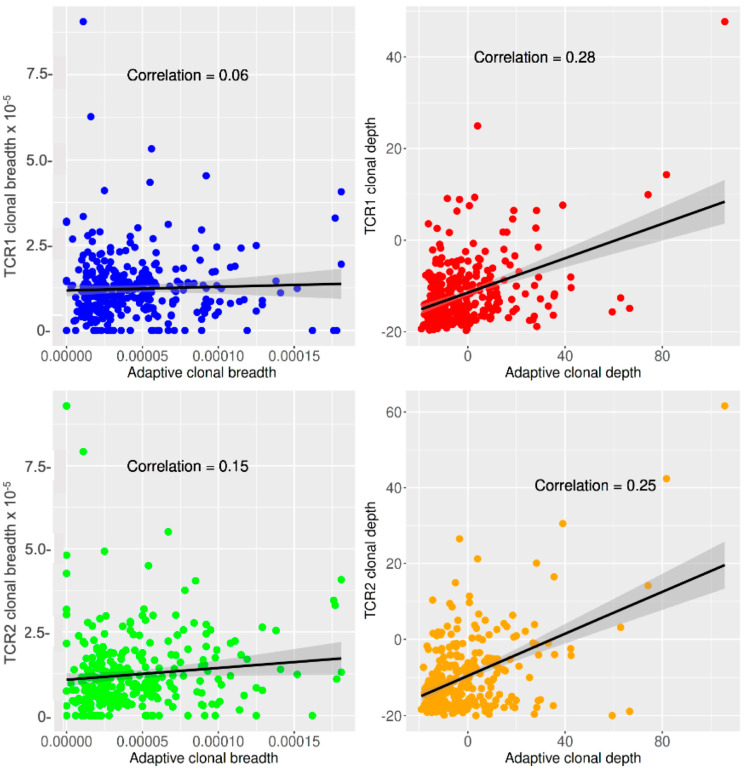
Correlation of private and public TCR vaccine-response metrics. **Upper panels**: TCR1 vs. public TCR (Adaptive) response metrics. **Lower panels**: TCR2 vs. Adaptive response metrics. **Left panels**: Clonal breadth. **Right panels**: Clonal depth.

**Figure 4 ijms-24-14223-f004:**
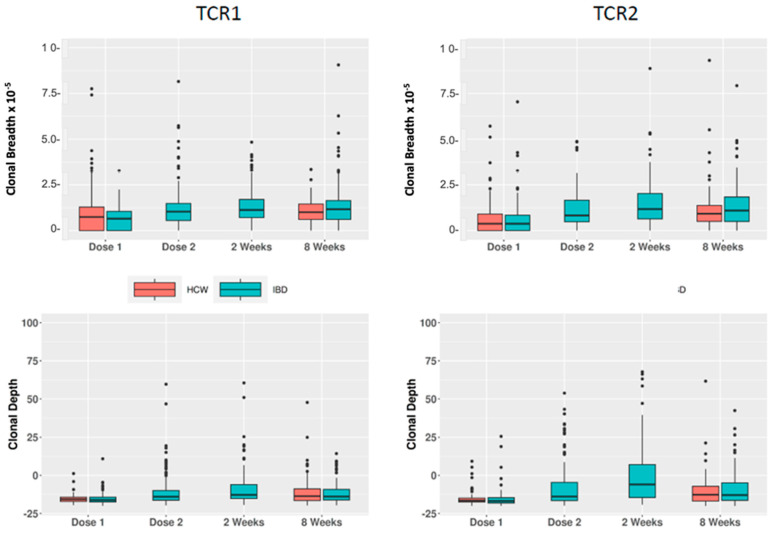
Time course of the private TCR vaccine response. **Upper panels***:* Clonal breadth. **Lower panels**: Clonal depth. **Left panels**: TCR1. **Right panels**: TCR2.

**Table 1 ijms-24-14223-t001:** Composition of the study cohorts.

Parameter	Dose 1	Dose 2	Week 2	Week 8	Total
Total subject number (%)	309 (33.33)	151 (16.29)	150 (16.18)	317 (34.2)	927
Cohort subject number (%)					
HCW	210 (67.96)	0 (0)	0 (0)	133 (41.96)	343 (37)
IBD	99 (32.04)	151 (100)	150 (100)	184 (58.04)	584 (63)
Age (median)	42	44	42	44	43
Sex (%)					
Male	198 (65.13)	82 (55.41)	80 (54.05)	198 (62.66)	558 (60.92)
Female	106 (34.87)	66 (44.59)	68 (45.95)	118 (37.34)	358 (39.08)
Vaccine type (%)					
BNT162 (Pfizer/BioNtech)	268 (89.04)	83 (54.97)	79 (55.63)	229 (75.33)	659 (73.39)
mRNA-1273 (Moderna/NIH)	33 (10.96)	68 (45.03)	63 (44.37)	75 (24.67)	239 (26.61)
Anti-TNF (%, IBD only)					
No	33 (33.33)	51 (33.77)	53 (35.33)	65 (35.33)	202 (34.59)
Yes	66 (66.67)	100 (66.23)	97 (64.67)	119 (64.67)	382 (65.41)
Any biologic (%, IBD only)					
No	24 (24.24)	37 (24.5)	32 (21.33)	44 (23.91)	137 (23.46)
Yes	75 (75.76)	114 (75.5)	118 (78.67)	140 (76.09)	447 (76.54)

**Table 2 ijms-24-14223-t002:** Comparison of private TCR response metrics at different intervals after vaccination in IBD and HCW cohorts.

Cohort	Comparison	Outcome	N	Estimate	Lower CI	Upper CI	*p*
IBD	week2 vs.	breadth.TCR1	214	0.25	0.18	0.32	8.90 × 10^−11^
	dose1	depth.TCR1	214	0.30	0.22	0.37	2.13 × 10^−13^
		breadth.TCR2	214	0.31	0.24	0.38	1.68 × 10^−15^
		depth.TCR2	214	0.41	0.34	0.48	2.21 × 10^−23^

IBD	week8 vs.	breadth.TCR1	242	0.26	0.19	0.33	1.06 × 10^−11^
	dose1	depth.TCR1	242	0.23	0.16	0.31	1.66 × 10^−9^
		breadth.TCR2	242	0.29	0.22	0.36	2.05 × 10^−14^
		depth.TCR2	242	0.31	0.25	0.38	2.45 × 10^−17^

HCW	week8 vs.	breadth.TCR1	321	0.12	0.06	0.19	2.01 × 10^−4^
	dose1	depth.TCR1	321	0.21	0.15	0.26	1.29 × 10^−12^
		breadth.TCR2	321	0.20	0.14	0.26	2.41 × 10^−10^
		depth.TCR2	321	0.21	0.16	0.26	1.40 × 10^−14^

Note: Among IBD subjects, samplings at dose1, week2, and week8 were variable. Accordingly, there is a difference in the N for subjects with available data for comparisons at week2 and week8. Analysis used a generalized linear model after inverse normal transformation.

**Table 3 ijms-24-14223-t003:** Age association of private TCR response metrics (8 weeks after vaccine).

Cohort	Outcome	N	Estimate	Lower CI	Upper CI	*p*
IBD	breadth.TCR1	163	−0.0023	−0.0052	0.0006	0.120
	depth.TCR1	163	−0.0054	−0.0080	−0.0027	9.69 × 10^−5^
	breadth.TCR2	163	−0.0007	−0.0037	0.0023	0.642
	depth.TCR2	163	−0.0040	−0.0068	−0.0012	5.51 × 10^−3^

HCW	breadth.TCR1	122	0.0000	−0.0034	0.0035	0.983
	depth.TCR1	122	−0.0033	−0.0069	0.0003	0.077
	breadth.TCR2	122	−0.0011	−0.0045	0.0023	0.539
	depth.TCR2	122	−0.0038	−0.0074	−0.0002	0.039

	breadth.TCR1	285	−0.0016	−0.0039	0.0006	0.149
Combined	depth.TCR1	285	−0.0047	−0.0068	−0.0025	2.71 × 10^−5^
	breadth.TCR2	285	−0.0009	−0.0031	0.0014	0.445
	depth.TCR2	285	−0.0039	−0.0061	−0.0017	5.59 × 10^−4^

Note: Shadows indicated highlighted findings.

**Table 4 ijms-24-14223-t004:** Effect of anti-TNF immunotherapy on the T-cell response.

Time Points	Outcome	Anti-TNF	No Biologic	Estimate	Lower CI	Upper CI	*p*
week2	breadth.TCR1	88	47	−0.04	−0.14	0.06	0.451
	depth.TCR1	88	47	0.12	0.03	0.22	0.014
	breadth.TCR2	88	47	−0.03	−0.14	0.07	0.539
	depth.TCR2	88	47	0.04	−0.06	0.15	0.426

week8	breadth.TCR1	103	60	0.05	−0.04	0.15	0.294
	depth.TCR1	103	60	0.08	−0.01	0.17	0.074
	breadth.TCR2	103	60	0.05	−0.05	0.15	0.361
	depth.TCR2	103	60	0.07	−0.02	0.16	0.130

Note: Comparisons are between anti-TNF and no anti-TNF (excluding patients on corticosteroids). Shadows highlight significant or trending comparisons.

## Data Availability

Requests for deidentified data may be directed to the corresponding author (J.B.) and will be reviewed by the Office of Research Administration at Cedars-Sinai Medical Center before issuance of data sharing agreements. Data limitations are designed to ensure patient and participant confidentiality.

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
