# Peer review of "A Paratope-Enhanced Method to Determine Breadth and Depth TCR Clonal Metrics of the Private Human T-Cell Vaccine Response after SARS-CoV-2 Vaccination"

_ijms, 2023, doi:10.3390/ijms241814223_

Round 1

Reviewer 1 Report

The manuscript presents a study on the effect of COVID-19 vaccine on TCR composition at different time-points using a bioinformatic approach taking advantage of the MIRA database of 130 000 putative SARS-CoV-2 specific TCRs. It is an interesting approach, but the work and its presentation have several flaws in its content and description. Specific comments are as follows:

1.     It is in general hard to follow the manuscript since a lot of important information is missing. For example, the concepts of clonal depth and clonal breadth should be clearly stated as well as a clearer definition of “adaptive” responses.

2.     Figure 1 should appear before table 1.

3.     Table 1, consider editing the table. I am not sure that data on patients on other biologicals than anti-TNF is included in the study?

4.     It should be clearly stated what model used in table 2-4. Estimate intercept as well as 5-95 CI of estimate should also be included in the table.

5.     It is also impossible to understand how the numbers add up in table 2? According to table 1 the number of participants sampled at 2 week is 150 and not 214 as stated in the table. Or do authors refer to the number of TCRs of interest after double filtration (n=251)? It is hard for the reader to tell since table 2 is not referred to in the text. Throughout the paper it is hard to tell what data the information in the tables are based on. Again, consider placing figures and tables in the order they are mentioned in the text.

6.     Table 3. Age association with what? More information is needed in the table. Also, in methods line 123 it is stated that linear model used for demographics and clinical features was adjusted for age and some other factors. I don’t really get it?

7.     Line 210, a reference is apparently missing.

8.     Figure 3. Add information to facilitate for the reader to understand and p-values should be included. It is not clearly stated how the correlation analyses were performed? What values were correlated? Rs=0.06 is hardly to be considered a correlation?

9.     Figure 4/Line 199. Authors state that they “…observed strong differences in the 199 CORALE cohort in both week2 vs baseline and week8 vs baseline comparison, for both groups of TCRs.” In figure 4 there are overlap in the IQRs at the different time-points and p-values are missing as well as info about what statistical test used, and number of values included in the testing. There are differences, but are they strong (significant)?

10.  Line 224 should read Discussion and after the discussion a conclusion should be made.

11.  Line 239, change the phrasing of the sentence for better understanding.

12.  Line 259-263. Authors claim that they show associations between vaccination timing, age and anti-TNF with the TCR response. They need to discuss more in detail how these conclusions can be made based on the findings of this study.

13.  Any impact on the results of the vaccine used and SARS-COV-2 infections should be included in the discussion.

Author Response

  1. It is in general hard to follow the manuscript since a lot of important information is missing. For example, the concepts of clonal depth and clonal breadth should be clearly stated as well as a clearer definition of “adaptive” responses.

Thanks for the helpful comments. We added the description and adaptive interpretations for breadth and depth metrics in the introduction and methods.

  1. Figure 1 should appear before table 1.

Thanks for this helpful comment. We made the suggested adjustment.

  1. Table 1, consider editing the table. I am not sure that data on patients on other biologicals than anti-TNF is included in the study?

Thanks for catching this. Patients on other biologicals were included in this study. We modified Table 1 to convey these details.

  1. It should be clearly stated what model used in Table 2-4. Estimate intercept as well as 5-95 CI of estimate should also be included in the table.

Thanks for pointing this out. We modified the results text to reflect the models used for Table 2-4. We have also added upper and lower 95% CI in the Tables. However, since it is not common practice to include or interpret intercepts of regression, we have not tabulated this parameter.

  1. It is also impossible to understand how the numbers add up in table 2? According to table 1 the number of participants sampled at 2 week is 150 and not 214 as stated in the table. Or do authors refer to the number of TCRs of interest after double filtration (n=251)? It is hard for the reader to tell since table 2 is not referred to in the text.

Thanks for catching this. For each CORALE subject, there were sometimes differences in the availability of sampling at the different time points, and hence differences in the N for the designated comparisons. This is now clarified in a Table 2 note. We also have now referred to Table 2 in the manuscript.

Throughout the paper it is hard to tell what data the information in the tables are based on. Again, consider placing figures and tables in the order they are mentioned in the text.

The results and table/figure order have been modified to clarify this issue.

  1. Table 3. Age association with what? More information is needed in the table. Also,in methods line 123 it is stated that linear model used for demographics and clinical features was adjusted for age and some other factors. I don’t really get it?

Thanks for catching this. We updated the title for Table 3 to clarify that subject age is being compared to private TCR response metrics. We have also revised the GLM description in methods to clarify the adjustment methodology.

  1. Line 210, a reference is apparently missing.

Thanks. The references are now added.

  1. Figure 3. Add information to facilitate for the reader to understand and p-values should be included. It is not clearly stated how the correlation analyses were performed?

What values were correlated? Rs=0.06 is hardly to be considered a correlation?

Thanks for the comments. We stated in the method section that the correlation was performed using spearman correlation. We now have also included p-values for the correlation in the text.

  1. Figure 4/Line 199. Authors state that they “…observed strong differences in the 199 CORALE cohort in both week2 vs baseline and week8 vs baseline comparison, for both groups of TCRs.” In figure 4 there are overlap in the IQRs at the different time-points and p-values are missing as well as info about what statistical test used, and number of

values included in the testing. There are differences, but are they strong (significant)?

Thanks to this comment. We have revised the text to indicate that Table 2 and Figure 4 are complementary presentations of findings on this point. Our revised description in the text details the findings and p-values. These indeed show a very strong statistical difference between time points.  

  1. Line 224 should read Discussion and after the discussion a conclusion should be made.

Thanks for the suggestion. We’ve edited the discussion title as suggested, and added a conclusion paragraph.

  1. Line 239, change the phrasing of the sentence for better understanding.

Thanks. We updated the phrasing accordingly.

  1. Line 259-263. Authors claim that they show associations between vaccination timing, age and anti-TNF with the TCR response. They need to discuss more in detail how these conclusions can be made based on the findings of this study.

Thanks for the suggestion. To address this, we have added comments and references to the relevant results text for the tables addressing these  points.

  1. Any impact on the results of the vaccine used and SARS-COV-2 infections should be included in the discussion.

Nearly all subjects were immunized with the Pfizer vaccine, and prior infection was <5%. So, the study was not powered to address these interesting questions.   

Reviewer 2 Report

The work of Dalin Li with co-authors is very important and necessary. The thing is that at present we have an arsenal of methods for assessing the humoral immune response, ranging from ELISA to virus neutralization tests. But methods for assessing the cellular response are not so accessible and convenient. There are a lot of difficulties with the analysis of the cellular response, so such work is very important.

The work was done at a high level. The necessary graphics are included. The conclusions are also justified, although the results themselves are rather weak, but since the topic is advanced, it is difficult to expect anything beyond.

As a small remark, it can be noted that there are errors in the text, for example, somewhere it is written SARS-CoV2 and somewhere SARS-CoV-2. It is necessary to carry out work to eliminate such errors.

Author Response

The work of Dalin Li with co-authors is very important and necessary. The thing is that at present we have an arsenal of methods for assessing the humoral immune response, ranging from ELISA to virus neutralization tests. But methods for assessing the cellular response are not so accessible and convenient. There are a lot of difficulties with the analysis of the cellular response, so such work is very important.

The work was done at a high level. The necessary graphics are included. The conclusions are also justified, although the results themselves are rather weak, but since the topic is advanced, it is difficult to expect anything beyond.

As a small remark, it can be noted that there are errors in the text, for example, somewhere it is written SARS-CoV2

We appreciate these helpful comments. We have identified and corrected a number of text errors (among which, the correction to SARS-CoV-2).   

Round 2

Reviewer 1 Report

The authors have addressed many of my main concerns about the manuscript but there are still some issues that authors need to adress: 

Authors should clearly state what p-values they considered significant. Usually p-values >0.05 are not considered significant. 

Line 47-48. Missing word? Consider rephrasing. 

Line 159 Table 1. 

“var” - spell out, not intuitive to the reader 

Left column first row, title is missing. Should read total or something like that? “N” as in numbers as well as % should be clearly indicated on each line. 

Line 178 – Something is off with the sentence. 

Line 186-188, Legend figure 2D. The numbers don’t add up. 4526+33=4559 and not 4599 as mentioned in legend and on line 194. In the figure it seems that as the number GLIPH CDR3 are 2834 and not 4599??? The characteristics of the additional filter for top 10 SOMs should also be mentioned in the legend to facilitate understanding. 

Line 209. r=0.06, p=0.09 is not a correlation, please consider rephrasing. 

Line 222. Change tempus of sentence. 

Line 234. Add information on what test used for association analyses and TCR metrics. It is easier to follow if estimate rather than beta is used on line 238, 239 and 244. 

Table 3 and Table 4, please consider to change the lay-out and add information to facilitate understanding. As I understand Spearman was used for correlation (line 142-142)? Spearman correlation is usually indicated by rs and should be included in the table. 

In Table 4 change “n0” and “n1” to “anti-TNF” and “no anti-TNF" or similar. It is also hard to understand what comparisons the results are based on. If you have compared with and w/o anti-TNF consider a lay-out similar to table 2, including headers “comparison” and “outcome. 

Table 1 and Table 4. In table 1 it seems as if the majority of patients in Corale have no anti-TNF, whereas in table 4 a majority of patients have anti-TNF. What is correct? And why is the total number of patients lower in table 4 compared to table 1. Selection criteria should be mentioned in methods/results and potential influence of the dropouts on the results should be discussed. 

Line 246-247 A graph of the comparison with-without anti-TNF will add value to the manuscript 

Line 282-283. Describe the results in fig 3 rather than referring to it in the text. 

A paragraph on limitations of the study should be included in the discussion.

Author Response

JB:  fix CORALE vs IBD

Comments to editor

A number of edits that we sent by email to *** on *** were not captured in the on-line version. So I have manually added them to the edited on-line version, as well as in the 2nd revision (termed v5).

Reviewer 2 items

The authors have addressed many of my main concerns about the manuscript but there are still some issues that authors need to adress: 

Authors should clearly state what p-values they considered significant. Usually p-values >0.05 are not considered significant. 

We also use that threshold. We have clarified in the text accordingly.

Line 47-48. Missing word? Consider rephrasing. 

We have rewritten this sentence for better clarity.

Line 159 Table 1. 

“var” - spell out, not intuitive to the reader 

Left column first row, title is missing. Should read “total” or something like that? “N” as in numbers as well as % should be clearly indicated on each line. 

Table 1 has been revised as suggested.

Line 178 – Something is off with the sentence. 

Revised.

Line 186-188, Legend figure 2D. The numbers don’t add up. 4526+33=4559 and not 4599 as mentioned in legend and on line 194. In the figure it seems that as the number GLIPH CDR3 are 2834 and not 4599??? The characteristics of the additional filter for top 10 SOMs should also be mentioned in the legend to facilitate understanding. 

Thanks for calling this out. The numbers all relate just to the GLIPH-filtered CDR3s. And, the total number (4599) refers to the total set of unique CDR3s identified by either the TOP3 and TOP10 parameters (as most of the smaller set of CDR3s (TOP10)- also appeared in the larger set (TOP3). We have edited the description to make it more clear.

Line 209. r=0.06, p=0.09 is not a correlation, please consider rephrasing. 

Rephrased.

Line 222. Change tempus of sentence. 

Rephrased.

Line 234. Add information on what test used for association analyses and TCR metrics. It is easier to follow if estimate rather than beta is used on line 238, 239 and 244. 

Done.

Table 3 and Table 4, please consider to change the lay-out and add information to facilitate understanding. As I understand Spearman was used for correlation (line 142-142)? Spearman correlation is usually indicated by “rs” and should be included in the table. 

Since all “r values” in the paper are Spearman, and this is clearly stated in the methods, we feel that it is unnecessary and misleading to use the subscript designation (this would prompt the statistics-minded reader to be on the lookout for different correlation methods among the presented data, which is not the case).

In Table 4 change “n0” and “n1” to “anti-TNF” and “no anti-TNF" or similar. It is also hard to understand what comparisons the results are based on. If you have compared with and w/o anti-TNF consider a lay-out similar to table 2, including headers “comparison” and “outcome”. 

Relabeling of Table 4 done. Comparison explanation is now highlighted as Table 4 note.

Table 1 and Table 4. In table 1 it seems as if the majority of patients in Corale have no anti-TNF, whereas in table 4 a majority of patients have anti-TNF. What is correct? And why is the total number of patients lower in table 4 compared to table 1. Selection criteria should be mentioned in methods/results and potential influence of the dropouts on the results should be discussed. 

There was a labeling error in Table 1 which is now corrected. The labeling of Table 4 is corrected (no anti-TNF is the correct descriptor. The explanation of number differences in Table 1 and 4 is due to the need to exclude patients on corticosteroid therapy, which reduces T-cell responses. This explanation is now added as a note in Table 4.

Line 246-247 A graph of the comparison with-without anti-TNF will add value to the manuscript 

We appreciate the suggestion but decline for the sake of timeliness.

Line 282-283. Describe the results in fig 3 rather than referring to it in the text. 

Done.

A paragraph on limitations of the study should be included in the discussion.

Done.